# Direct visualization of critical hydrogen atoms in a pyridoxal 5′-phosphate enzyme

Steven Dajnowicz[1,2], Ryne C. Johnston [3], Jerry M. Parks [3], Matthew P. Blakeley [4], David A. Keen [5], Kevin L. Weiss [2], Oksana Gerlits[6], Andrey Kovalevsky [2] & Timothy C. Mueser[1]

Enzymes dependent on pyridoxal 5′-phosphate (PLP, the active form of vitamin $B_6$) perform a myriad of diverse chemical transformations. They promote various reactions by modulating the electronic states of PLP through weak interactions in the active site. Neutron crystallography has the unique ability of visualizing the nuclear positions of hydrogen atoms in macromolecules. Here we present a room-temperature neutron structure of a homodimeric PLP-dependent enzyme, aspartate aminotransferase, which was reacted in situ with α-methylaspartate. In one monomer, the PLP remained as an internal aldimine with a deprotonated Schiff base. In the second monomer, the external aldimine formed with the substrate analog. We observe a deuterium equidistant between the Schiff base and the C-terminal carboxylate of the substrate, a position indicative of a low-barrier hydrogen bond. Quantum chemical calculations and a low-pH room-temperature X-ray structure provide insight into the physical phenomena that control the electronic modulation in aspartate aminotransferase.

[1] Department of Chemistry and Biochemistry, University of Toledo, Toledo, OH 43606, USA. [2] Biology and Soft Matter Division, Oak Ridge National Laboratory, Oak Ridge, TN 37830, USA. [3] UT/ORNL Center for Molecular Biophysics, Biosciences Division, Oak Ridge National Laboratory, Oak Ridge, TN 37830, USA. [4] Large-Scale Structures Group, Institut Laue Langevin, Grenoble Cedex 9 38042, France. [5] ISIS Facility, Rutherford Appleton Laboratory, Harwell Campus, Didcot OX11 0QX, UK. [6] UT/ORNL Joint Institute of Biological Sciences, University of Tennessee, Knoxville, TN 37830, USA. Correspondence and requests for materials should be addressed to A.K. (email: kovalevskyay@ornl.gov) or to T.C.M. (email: timothy.mueser@utoledo.edu)

The vitamin B$_6$ complex, consisting of pyridoxine, pyridoxal, pyridoxamine, and their phosphorylated derivatives, is involved in neurotransmitter synthesis, amino acid metabolism, glycogen metabolism, and other physiological pathways[1]. Pyridoxal 5′-phosphate (PLP), the biologically active cofactor derived from pyridoxine, is one of the most ubiquitous cofactors found in nature, catalyzing ~140 different types of biochemical transformations[1–4]. PLP is used in transamination, racemization, phosphorylation, decarboxylation, aldol cleavage, elimination, and replacement reactions[1–3]. PLP-dependent enzymes are categorized into five recognized fold-types, each performing characteristic chemistry[1, 2]. Such diverse chemical reactions found in PLP-dependent enzymes have intrigued researchers for decades. Fold-type I is the most prevalent, mainly promoting transamination and decarboxylation reactions[1]. Two major hypotheses have been developed to understand the different types of chemistry in PLP-dependent enzymes, namely (1) stereoelectronic control and (2) electronic modulation through selective protonation.

Stereoelectronic control of PLP, also known as the Dunathan hypothesis, involves substrate destabilization by orienting the reactive bond perpendicular to the conjugated π-system in PLP[5]. Alternatively, the hypothesis of electronic modulation through selective protonation asserts that the local active site environment of the enzyme promotes specific protonation states of the cofactor, suggesting that proton positions govern reaction diversity. Thus, the active site protonation states or "protonation profiles" of PLP-dependent enzymes may serve to optimize the desired chemical transformation[2]. The different fold-types of PLP-dependent enzymes induce specific protonation profiles, promoting specific reactions while prohibiting chemically viable side reactions. Verification of the latter hypothesis has been the focus of several NMR studies[6–9], but remains elusive. Recent work with NMR crystallography has determined the protonation profiles of PLP in multiple states for tryptophan synthase[6], a fold-type II PLP-dependent enzyme that promotes β-elimination.

Neutron crystallography is uniquely able to determine all the positions of hydrogen/deuterium (H/D) atoms within a protein[10]. From the coherent nuclear scattering, the nuclear positions of H/D can be directly determined at high to medium resolutions (2.5 Å and higher). Here we report the neutron crystal structure of a PLP-dependent enzyme, aspartate aminotransferase (AAT), a fold-type I PLP-dependent enzyme that reversibly converts L-aspartate and α-ketoglutarate to oxaloacetate and L-glutamate via a ping–pong bi–bi mechanism[3].

Recombinant porcine cytosolic AAT fortuitously packs as a biological and crystallographic homodimer, with the two active sites having different activities toward the substrate analog α-methylaspartic acid[11, 12]. Thus, in the room-temperature neutron structure of AAT (Fig. 1), after the crystal was soaked with α-methylaspartic acid, one site reacts and closes (chain A), whereas the second one remains unreacted and open (chain B). In the open monomer, the Schiff base (SB) is in the internal aldimine form and PLP is covalently linked to a conserved lysine residue (K258). The closed monomer displays the SB in the external aldimine form, with PLP covalently linked to the Cα of α-methylaspartate. Obtaining both states in one crystal enables a direct comparison of the H/D positions in each state[11, 12] (Fig. 1). To help decipher the impact of protonation, we also determined a low pH, room-temperature X-ray structure of AAT. At pH 4.0, the internal aldimine shows geometric differences near the SB of PLP compared to the neutron structure at pH 7.5. Forced protonation alters the PLP alignment, suggesting that the microenvironment involving Y225 and the phenolic oxygen (O3′) of PLP influence the SB protonation and reactivity. Quantum chemical calculations support the inherent non-coplanarity of the deprotonated SB nitrogen (N$_{SB}$) in the internal aldimine observed in the neutron structure, and suggest that this geometry is stabilized by hyperconjugation, instead of being destabilized by previously proposed strain[13] involving K258. Moreover, our structural analysis provides insight into how PLP enzymes selectively protonate cofactors, substrates, or both during the

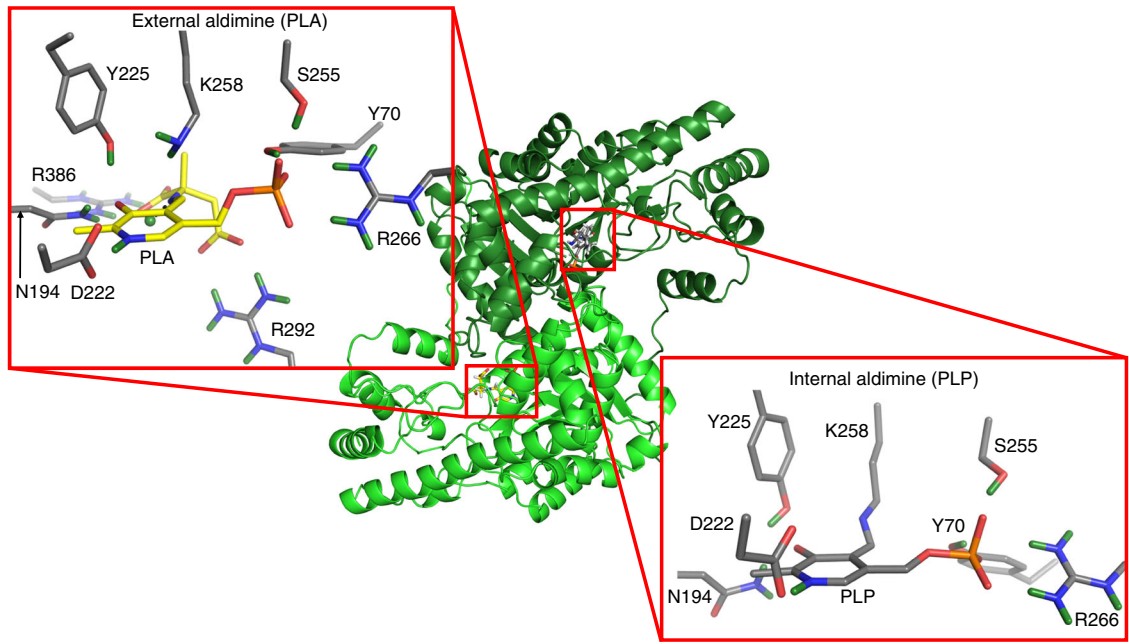

**Fig. 1** Structure of AAT with α-methylaspartate. The PLP cofactor in the external aldimine (chain A in light green) and the internal aldimine (chain B in dark green) forms are shown in yellow and gray carbon scheme, respectively. In the external aldimine, R292 and R386 interact directly with the carboxylate groups of α-methylaspartate. These interactions are not present in the internal aldimine. Here and in Figs. 2, 3, and 5, the deuterium atoms are colored green and the external aldimine is labeled as PLA

catalytic cycle to promote the desired electronic configurations for specific chemical reactions.

## Results

**Neutron diffraction.** The neutron structure of AAT in complex with α-methylaspartic acid was solved to 2.2 Å resolution (PDB ID 5VJZ) at room temperature (Table 1). Deuterated recombinant AAT was purified and crystallized using hydrogenated reagents. The crystals were exchanged by vapour diffusion with $D_2O$ for an extensive period of time (>1 yr), replacing exchangeable hydrogens with deuterium. The crystal structure contains both the internal (chain B) and external (chain A) aldimines in the homodimer present in one asymmetric unit. It appears that the two different states were obtained because crystal contacts prohibit one monomer from closing, which is necessary for the enzyme to stabilize a Michaelis complex (Supplementary

Fig. 1). The cofactor visualized in the active sites is a mixture of deuterated endogenous PLP and hydrogenated PLP added during crystallization. In both active sites, Y70 side chains act as bridges to the opposing active site, making tight 2.4–2.5 Å hydrogen bonds with the PLP phosphate moieties. R266 (in chains A and B) is cationic, and in a salt bridge with the PLP phosphate. R292(B) and R386(A) are also cationic as they interact with the substrate in the external aldimine.

For the internal aldimine (chain B), no nuclear density peaks for deuterium were observed near the O3′ or SB of the PLP cofactor (Fig. 2a). It has been suggested previously that the $pK_a$ of the $N_{SB}$ is ~6.5 for AAT, but estimates vary in the literature (5.4–7.0)[13–15]. The lack of neutron density for the internal aldimine O3′ deuteron suggests a deprotonated state, which would be stabilized by the PLP resonance to give the C–O⁻ bond a partial double bond character (Fig. 2b). Y225 forms a close hydrogen bond with PLP O3′ (2.6 Å), indicative of a strong H bond. The side-chain amide of N194 forms a close H bond with O3′ as well. More importantly, the SB C=N double bond is 46° above the plane of the pyridinium ring, which indicates the lack of a hydrogen bond to O3′. The out-of-plane geometry for the SB observed in our neutron structure was also identified in other aminotransferase enzymes, with torsion angles ranging from 43 to 96°[13]. Overall, the experimental evidence demonstrates that O3′ and the SB are deprotonated and likely carry partial negative charges, stabilized by PLP resonance and H bonds with Y225 and N194.

In the external aldimine (chain A), the Y225 hydroxyl adopts the position of a hydrogen bond donor to the PLP O3′ (2.9 Å) with O3′ deprotonated similar to the internal aldimine (Fig. 3a). Unaltered, N194 remains as a close H bond donor to O3′. Additionally, K258, which forms the SB in the internal aldimine, is neutral in the external aldimine active site (Fig. 3c). The newly formed SB in the external aldimine adopts a −28° dihedral angle, positioning the $N_{SB}$ below the PLP pyridinium ring, a 74° difference compared with the internal aldimine geometry. A strong nuclear scattering length density peak for a deuterium appears near the $N_{SB}$ in the external aldimine (chain A), indicating that the SB is protonated (Fig. 3a). Remarkably, the peak is not between O3′ and $N_{SB}$, but is instead located between $N_{SB}$ and the C-terminal carboxylate oxygen of α-methylaspartate (Fig. 3a). The N···O distance is 2.6 Å and the deuteron is equidistant between the two heteroatoms, with N···D and O···D distances of 1.5 Å (Fig. 3). The O3′···D distance is 2.5 Å, which is too long to be considered a hydrogen bond, and therefore it cannot contribute to the planarity of the SB and PLP. Thus, the difference in the SB out-of-plane geometries is likely due to changes in the electronics between the internal and external aldimine states and not because of hydrogen bond formation between O3′ and $N_{SB}$.

In AAT, the protonated pyridine nitrogen of the PLP (N1-PLP) enhances charge delocalization through resonance, permitting the subsequent transamination reaction (Fig. 4)[3, 16]. The neutron structure reveals that the N1-PLP is protonated in both the internal and external aldimines (Figs. 2, 3), consistent with previous NMR studies[17]. In neutron protein crystallography the levels of H/D exchange can be estimated by refining occupancies of the D atoms[18] (see Methods section for details). Interestingly, the D atoms on N1-PLP display different levels of exchange in the two states. In the internal aldimine, the N1-PLP hydrogen is essentially fully exchanged with D (D occupancy = 0.91), whereas in the external aldimine it is only partially exchanged (D occupancy = 0.45). Thus, the rate of H/D exchange of N1-PLP is reduced in the external aldimine state. The observation of the partially exchanged D on N1-PLP in the external aldimine may indicate a stronger and less mobile hydrogen bond with D222 than in the internal aldimine.

| Table 1 X-ray and neutron crystallographic data collection and refinement | | |
|---|---|---|
| | **pH 7.5** | **pH 4.0** |
| | **PDB 5VJZ** | **PDB 5VK7** |
| *Data collection (neutron)* | | |
| Beamline/facility | LADI-III/ILL | |
| Resolution range (Å) | 55.19–2.20 | |
| | (2.32–2.20) | |
| Space group | $P2_12_12_1$ | |
| Cell dimensions | | |
| *a, b, c* (Å) | 55.53, 123.82, | |
| | 129.73 | |
| α, β, γ (°) | 90, 90, 90 | |
| No. reflections total | 96,244 (9488) | |
| No. reflections unique | 32,950 (3950) | |
| Completeness (%) | 72.9 (60.4) | |
| *I/σI* | 6.5 (4.9) | |
| $R_{merge}$ | 0.142 (0.186) | |
| Redundancy | 2.9 (2.4) | |
| *Data collection (X-ray)* | | |
| Beamline/facility | Rigaku HighFlux | Rigaku HighFlux |
| | Home-Source | Home-Source |
| Resolution range (Å) | 50.00–2.00 | 40.00–1.9 |
| | (2.07–2.00) | (1.97–1.90) |
| No. reflections unique | 62,791 (5739) | 69,048 (7027) |
| Completeness (%) | 95.5 (92.9) | 94.4 (97.9) |
| *I/σI* | 28.3 (5.2) | 24.2 (3.4) |
| $R_{merge}$ | 0.031 (0.470) | 0.052 (0.376) |
| Redundancy | 6.3 (6.3) | 4.6 (4.6) |
| *Joint X/N refinement* | | *X-ray refinement* |
| Resolution (neutron, Å) | 20–2.20 | n/a |
| Resolution (X-ray, Å) | 20–2.0 | 30–1.9 |
| Data rejection criteria | No observation | n/a |
| | and \|F\| = 0 | |
| Sigma cutoff | 2.5 | n/a |
| No. reflections (neutron) | 32,558 | n/a |
| No. reflections (X-ray) | 55,493 | 64,425 |
| $R_{work}/R_{free}$ (neutron) | 0.234/0.256 | n/a |
| $R_{work}/R_{free}$ (X-ray) | 0.197/0.220 | 0.185/0.225 |
| No. of atoms | | |
| Protein including H and D | 12,978 | 6429 |
| Cofactor/ligand | 81 | 48 |
| Water | 735 (245 $D_2O$) | 232 (232 O atoms) |
| R.m.s deviations | | |
| Bond lengths (Å) | 0.008 | 0.006 |
| Bond angles (°) | 1.04 | 1.02 |
| Average *B*-factor | | |
| Protein | 28.7 | 31.4 |
| Cofactor/ligand | 27.3 | 17.1 |
| Water | 39.7 | 35.2 |

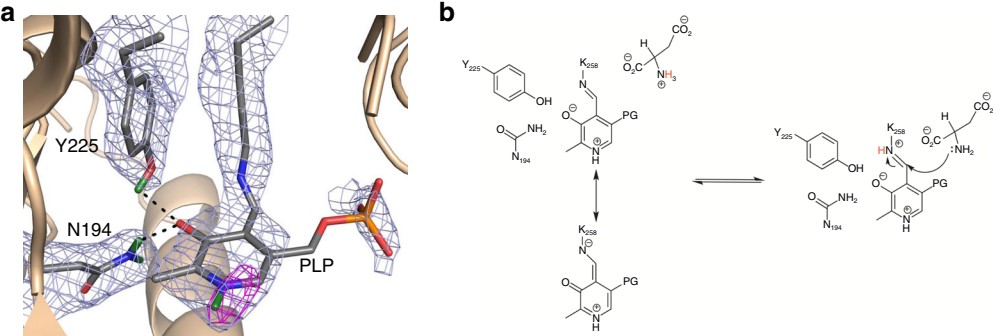

**Fig. 2** Internal aldimine reactivity and substrate activation. **a** Internal aldimine state in the AAT-α-methylaspartate neutron structure. Here and in Figs. 3 and 5, the light blue mesh is the 2|$F_o$|–|$F_c$| nuclear scattering length density at 1σ and the magenta mesh is the omit |$F_o$|–|$F_c$| difference nuclear scattering length density at 3σ. For clarity, the omit |$F_o$|–|$F_c$| difference nuclear scattering maps are presented for deuterium atoms on PLP. The dashed lines indicate hydrogen bonds. **b** Michaelis complex and proposed substrate activation with the proton transferred upon substrate binding shown in red. A direct or indirect proton transfer mechanism is plausible. Two possible resonance structures of the internal aldimine are shown

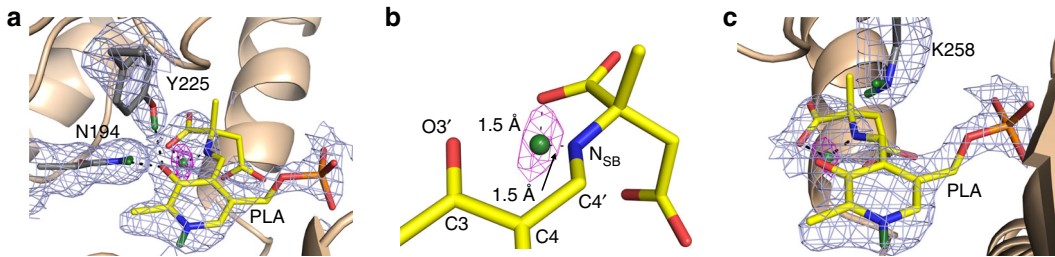

**Fig. 3** Neutron structure of the external aldimine. **a** External aldimine state in the AAT-α-methylaspartate neutron structure. **b** Close-up of the D atom that is equidistant between the $N_{SB}$ and carboxylate oxygen. **c** External aldimine and the catalytic base, K258. For clarity, the omit |$F_o$|–|$F_c$| difference nuclear scattering map is presented for the deuterium atom on PLA

The H bond network consisting of D222, H143, T139, H189, and a cluster of water molecules (Supplementary Fig. 2) is coupled from N1-PLP to the bulk solvent. This H bond network significantly increases the p$K_a$ of N1-PLP (>7.0) in the active site of AAT compared to 5.8 in solution[19]. Moreover, in the internal aldimine, H143 and H189 are neutral, while in the external aldimine H143 remains neutral but H189 becomes protonated and positively charged (Fig. 5). Our neutron structure agrees with [1]H NMR experiments[20–22] and quantum chemical cluster models of the active site[12], which showed that H143 and H189 are uncharged (singly protonated) in the internal aldimine state. In the external aldimine, H189 becomes doubly protonated. Furthermore, the electrostatic contribution of H189 was shown to affect the rate of catalysis in AAT, with a 75% decrease in $k_{cat}$ for the H143L:H189L double mutant compared with a 50% decrease for the H143L single mutant[12]. Thus, as more electron density is added in the external aldimine state, additional positive charge is perhaps necessary to counterbalance the negative charges from the carboxylic groups of the substrate. Due to H189 protonation, the water molecule ($D_2O$) that is H bonded to $N^{\delta 1}$ of H189 changes its relative orientation in the two states (Fig. 5). A cluster of three water molecules ($D_2O$) coupled to H189 through H bonds (Supplementary Fig. 2) leads to the bulk solvent through a second cluster of water molecules ($D_2O$). The water molecules in the second cluster are not directly connected by H bonds to those in the first cluster, but are more mobile with high B-factors. Thus, they can rearrange to promote a proton transfer by the Grotthüss proton hopping mechanism to shuttle protons in and out of the active site (Fig. 5c). This water channel is perhaps responsible for H189 protonation when the external aldimine is

formed and may also assist in N1-PLP protonation when the apoenzyme reacts with PLP to generate the internal aldimine.

**Low-pH X-ray structure.** To probe the structural changes that occur upon internal aldimine SB protonation, we obtained a low-pH X-ray structure of AAT in the absence of substrate, in which both chains are in the internal aldimine state. Acidifying AAT crystals with acetic acid vapour, using a procedure described previously[23, 24], yielded a 1.9 Å room temperature X-ray structure at pH ~4.0 (PDB ID 5VK7; Table 1). When the geometries of the internal aldimines (chain B) from the pH 7.5 and pH 4.0 structures were compared, two major differences were evident. The torsion angle between the SB C=N bond and the pyridinium ring, and the O···O distance between the Y225 hydroxyl and O3′ of PLP are different (Fig. 6). In the pH 4.0 structure, the SB is 22° above the pyridinium ring plane, compared to 46° in the internal aldimine at pH 7.5 (Fig. 6). Additionally, in the pH 4.0 structure the O···O distance between the Y225 hydroxyl and O3′ of PLP is 2.9 Å, compared with 2.6 Å in the internal aldimine state at pH 7.5 (Fig. 6). In the other monomer of the pH 4.0 structure (chain A, with restricted motion), the corresponding torsion angle and the (Y225)O···O(O3′) distance are 26° and 3.0 Å, respectively. Protonation of the SB nitrogen of the internal aldimine at low pH appears to introduce an H bond between O3′ and the SB by reducing the torsion angle between SB and the pyridinium ring. However, protonation of O3′ cannot be ruled out as H atoms were not observed in the low pH X-ray structure.

**Quantum chemical calculations.** Previous studies have suggested that the out-of-plane geometry of the internal aldimine SB is

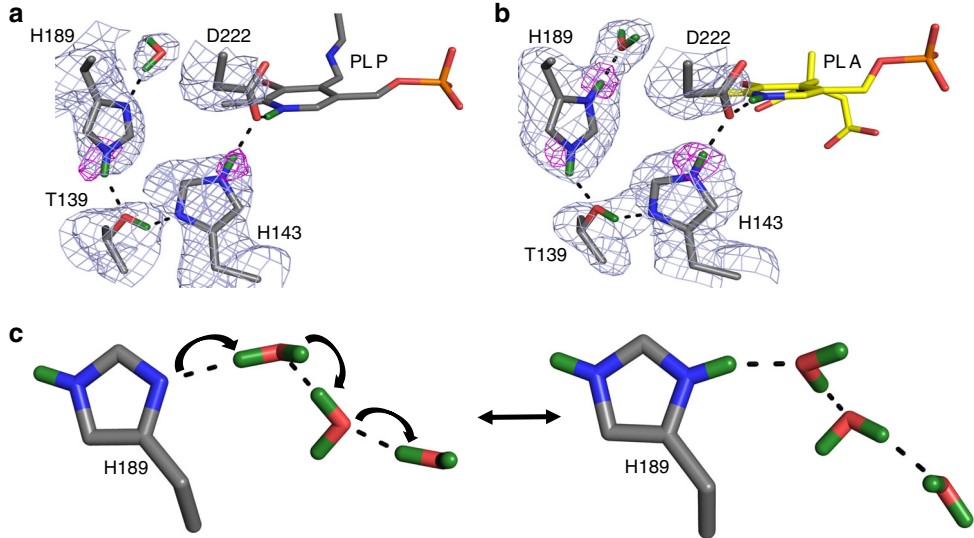

**Fig. 4** Proposed mechanism for the first half-reaction of AAT. The mechanism starts from the external aldimine obtained by substrate attack on the internal aldimine, as shown in Fig. 2b

External aldimine Carbanionic intermediate Quinonoid intermediate Pyridoxamine 5′-phosphate + oxaloacetate

**Fig. 5** Extended hydrogen bond network near N1-PLP in the internal aldimine **a** and external aldimine **b**. **c** Proposed Grotthüss proton hopping mechanism, leading to the protonation of H189 during the formation of the external aldimine. For clarity, the omit $|F_o|-|F_c|$ difference nuclear scattering maps are presented for deuterium atoms on His residues

induced by strain caused by the side chain of K258. This strain was proposed to cause ground state destabilization of the cofactor, thus contributing to the "catalytic power" of the enzyme[13]. To probe the nature of this geometric distortion intrinsic to the internal aldimine intramolecular interactions, we performed density functional theory (DFT) calculations using a truncated Lys-PLP model in isolation from the active site (Methods section). Using the protonation states observed in the neutron structure, unconstrained geometry optimization of the internal aldimine model resulted in a C3–C4–C4′– $N_{SB}$ torsion angle of 42°, in excellent agreement with the neutron structure. This finding suggests that the non-coplanarity of the two PLP moieties cannot be explained solely by strain, and that intramolecular electronic forces play an important role. Upon $N_{SB}$ or O3′ protonation, the SB-to-pyridinium torsion angle becomes essentially 0° (Supplementary Fig. 3), resulting from the presence of the incipient $N_{SB}$···O3′ hydrogen bond. This finding is consistent with the reduced torsion angle of 22° in the pH 4.0 X-ray structure, in which we expect $N_{SB}$ to be protonated. The deviation in the torsion angle between the DFT model and the low-pH structure can be attributed to geometric restraints imposed by the active site that are not present in the DFT models. For example, the π–π stacking interaction between the pyridine ring of PLP and W140 were excluded in the simplified model. Nevertheless, the intramolecular orbital interactions can be considered primary (first-order) effects, whereas noncovalent interactions with nearby residues are second-order. Interestingly, when the N1-PLP and $N_{SB}$ are both deprotonated in the DFT model, the SB double bond is coplanar (3°) with neutral pyridine (Supplementary Fig. 3), suggesting that the electron-withdrawing capabilities of the pyridine ring influence the SB torsion angle.

Using natural bond orbital analysis, we found that the distortion from planarity arises from electronic effects that include electron repulsion between the $N_{SB}$ and O3′ lone pairs, and hyperconjugation (Fig. 7). Hyperconjugation is a stabilizing interaction in which the electrons in either a σ or π-bond are delocalized into an empty or partially filled antibonding σ or π-orbital (σ* or π*). The π → π* interaction energies of the $\pi_{SB'} \to \pi^*_{C3-C4}$ and $\pi_{C3-C4} \to \pi^*_{SB'}$ are 5.2 and 12.3 kcal mol$^{-1}$, respectively (Fig. 7), stabilizing the out-of-plane conformation by 17.5 kcal mol$^{-1}$. The presence of the hyperconjugative interactions offsets the disruption of conjugation caused by lone-pair repulsion. Similar interactions are present in biphenyl, which has maximal hyperconjugation interactions of ~8.0 kcal mol$^{-1}$ and a phenyl-to-phenyl torsion angle of 44°[25, 26]. Thus, the interplay of favorable conjugative and hyperconjugative interactions in the internal aldimine contribute to offsetting the lone-pair repulsion (Fig. 7), allowing the SB to adopt the observed non-

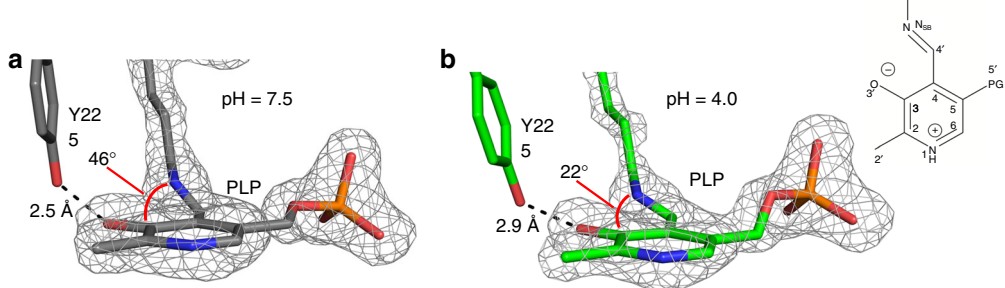

**Fig. 6** X-ray structures of AAT at different pH. Internal aldimine at pH 7.5 **a** and pH 4.0 **b**. The gray mesh is the omit |F$_o$|−|F$_c$| difference electron density at 5σ

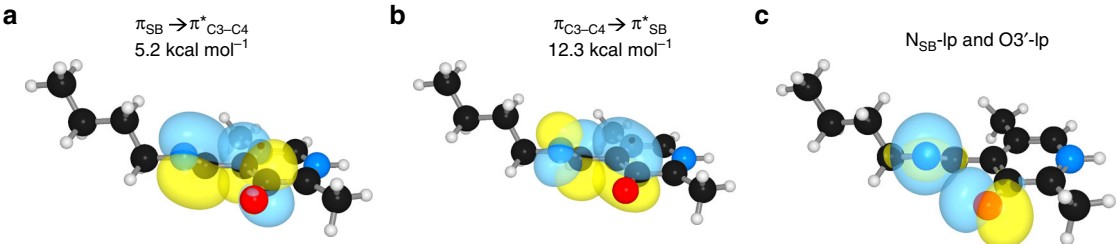

**Fig. 7** Natural bond orbitals in the N$_{SB}$-deprotonated/N1-protonated internal aldimine model. **a** π$_{SB-C4'}$ → π*$_{C3-C4}$, **b** π$_{C3-C4}$ → π*$_{SB-C4'}$, and **c** lone-pair (lp) interactions

coplanar structure. In the protonated SB model, hyperconjugative interactions between N$_{SB}$–C4′ and C3–C4 significantly decrease, to <0.5 kcal mol$^{-1}$. A strong hydrogen bond forms between N$_{SB}$ and O3′, which allows adoption of a more planar structure. When the N1-PLP is deprotonated the π–π conjugation must increase, stabilizing the SB to be coplanar with the ring despite unfavorable lone-pair repulsion. In both the internal and external aldimines, the SB planarity and reactivity can be tuned by PLP-N1 protonation, a demonstration of electronic modulation by selective protonation. All donor−acceptor orbital interactions were computed with a threshold of 0.5 kcal mol$^{-1}$ [27, 28] (Supplemental Information).

## Discussion

The diverse chemistry performed by PLP-dependent enzymes has been poorly understood and was only recently revised through insight from NMR studies[6–9]. Historically, many of the protonation states of PLP and active site residues were assigned based on spectral analysis or chemical intuition from X-ray structures. The determination of the protonation states by either NMR or neutron crystallography has the power to answer many questions in enzymology and structural biology.

The proposed mechanisms of PLP-catalyzed transamination reactions require a neutral α-amine on the substrate for external aldimine formation. The neutron structure shows that the internal aldimine N$_{SB}$ is deprotonated near-physiological pH (Fig. 2). Therefore, the amino acid substrate may bind initially as a zwitterion in the Michaelis complex and the internal aldimine nitrogen could then accept a proton from the N-terminal NH$_3^+$ of the substrate, thereby activating the cofactor (Fig. 2b). The bond between PLP and the conserved lysine is cleaved during the formation of the external aldimine (Fig. 4). We found a deuteron positioned equidistant between the N$_{SB}$ nitrogen and oxygen of the substrate C terminus oxygen in the external aldimine, indicative of a low-barrier hydrogen bond (LBHB). In an LBHB, the

proton can move back and forth between the two heavy atoms because the zero-point vibrational energy is above the potential energy barrier[29, 30]. Here, the formation of the intramolecular LBHB between the N$_{SB}$ and carboxylate oxygen was unexpected. Historically, a double-well hydrogen bond would be expected to have formed between the N$_{SB}$ and O3′ of PLP[3].

Previous quantum mechanics/molecular mechanics (QM/MM) calculations suggested that the tautomeric equilibrium between the N$_{SB}$ and O3′ in AAT is shifted toward the N$_{SB}$ by >7.0 kcal mol$^{-1}$[31]. However, to our knowledge, the LBHB (Fig. 3) observed in our neutron structure in the external aldimine has not been detected or proposed for any PLP-dependent enzyme. Interestingly, previous studies have suggested that if the carboxylate oxygen, O3′, or both are protonated in PLP-glycine adducts, the p$K_a$ of the Cα proton would be significantly decreased, making it more acidic[4, 32]. The D atom present in the LBHB in our neutron structure would be highly polarized, containing an increased partial positive charge. Such intramolecular hydrogen bonding may also be present in other PLP-dependent enzymes, in which stabilization of the carbanion (an azomethine ylide) is needed for specific chemistry. The presence of the LBHB is further supported by the SB being in the plane of the carboxylate oxygen, making the SB go 28° below the plane of the pyridinium ring rather than being coplanar with O3′.

In the external aldimine the ε-amine of K258 has been proposed to act as a general base that abstracts the Cα proton from the external aldimine, leading to the formation of a carbanionic intermediate (Fig. 4). The Cα deprotonation is partially rate-limiting in AAT[1–3]. In the case of the α-methylaspartate substrate used in this study, the reactive H on Cα of aspartate has been substituted with a methyl group, precluding the advancement of the AAT-catalyzed reaction beyond the external aldimine state. The neutron structure shows that K258 is neutral (ND$_2$, Fig. 3c), corroborating its role as the general base catalyst[33]. Thus, the proton inventory remains constant between the internal and

external aldimines in our neutron structure: two protons from the protonated α-amino group of the substrate are transferred to K258 and one proton remains on the nascent $N_{SB}$ when the external aldimine is formed.

A recent NMR study has shown that the internal aldimine of tryptophan synthase has a protonated $N_{SB}$ and a deprotonated N1-PLP[34]. In AAT, N1-PLP is protonated in both the internal and external aldimine forms. AAT catalyzes transaminations, whereas tryptophan synthase performs β-eliminations. The difference in the local active site environments of AAT and tryptophan synthase is responsible for the different observed protonation profiles. Specifically, in AAT, an Asp residue forms a salt bridge with N1-PLP, while in tryptophan synthase a Ser residue is hydrogen bonded to N1-PLP. The different protonation states observed in the two enzymes may be responsible for promoting these two specific types of chemistries and preventing side reactions.

We now summarize our observations and calculations as they apply to the transamination reaction as follows:

(1) We propose that when the apo-enzyme reacts with PLP to form the internal aldimine, the hydrogen bond network spanning from N1-PLP to the bulk solvent through D222, H143, T139, and H189 is responsible for the N1-PLP protonation via the Grotthüss proton hopping mechanism (Fig. 5c)[12]. N1-PLP protonation influences the coplanarity of the SB relative to the pyridinium ring, with the SB C=N torsion angle tilted by 46°. In addition, N1-PLP protonation is important for the enhanced electron sink effect for stabilization of the carbanion intermediate.

(2) O3′ is not protonated in either the internal or external aldimines at near-physiological pH. The negative charge on O3′ is stabilized by resonance, leading to partial negative charges on O3′ and the $N_{SB}$ in the internal aldimine. In both aldimines the negative charge on O3′ is further stabilized by hydrogen bonds with Y225 and N194.

(3) $N_{SB}$ is not protonated in the internal aldimine, but is ready to accept a proton from the incoming N-terminal $NH_3^+$ group of the substrate aspartate. Deprotonation of the substrate $NH_3^+$ activates it to attack the SB carbon and proceed to external aldimine formation. As the protonated $N_{SB}$ switches linkage from K258 to the substrate, the SB C=N bond rotates below (−28°) the pyridinium ring to H bond with the C-terminal carboxylate of the substrate.

(4) In the external aldimine, a deuterium is positioned midway between $N_{SB}$ and the C-terminal carboxylic oxygen of α-methylaspartate, participating in an apparent LBHB. This LBHB may stabilize the subsequent formation of the carbanion after Cα is deprotonated by K258 by "shielding" the negative charge of the carboxylate from the Cα carbon.

(5) H189 is neutral in the internal aldimine, but becomes protonated, presumably through the water channel, in the external aldimine. The resultant extra positive charge is an additional counterbalance to the negative charge on Cα in the ensuing carbanion.

The direct determination of the hydrogen positions in AAT explains how PLP-dependent enzymes promote specific protonation states and change hydrogen bonding configurations between intermediate steps in the transamination catalytic cycle. As conformational and electrostatic changes are invoked during cofactor and substrate binding, several proton transfers occur to modulate the required electronic configurations. Nature has fine-tuned biochemical reactions by stereoelectronic control and electronic modulation through selective protonation within the active sites of PLP-dependent enzymes.

## Methods

**Protein purification and crystallization**. For deuterated protein expression of AAT, *Escherichia coli* T7 Express Lac[IQ] cells (New England BioLabs) were grown in minimal medium with 100% $D_2O$ and hydrogenous glycerol as the sole carbon source. A bioreactor (BIOFLO 310, Eppendorf) was used for protein expression to obtain high quantities of deuterated protein. The cells were grown to an $OD_{600}$ of 10.0 at 30 °C, induced with 1 mM isopropyl β-D-thiogalactopyranoside and fed with hydrogenous glycerol for 24 h at 20 °C. Cell pastes were stored at −80 °C until needed for protein purification. Purification and of AAT is performed by using a Ni-NTA affinity chromatography procedure[12]. The batch method was used for crystallization (crystallization solution contained 40 mM NaOAc (pH 5.4), 2 mM PLP, 9% w/v polyethylene glycol (PEG) 6000, and 10% glycerol). To obtain AAT in complex with α-methylaspartate, crystals were soaked in a new solution containing 50 mM Tris-HCl (pH 7.5), 5% w/v PEG 6000, 5% glycerol, 300 mM α-methyl-DL-aspartate. Crystals suitable for neutron diffraction (0.5–0.65 mm³) were then mounted in capillaries that contained the same soaking solution but in $D_2O$ to exchange any labile hydrogens with deuterium. The crystal underwent H/D exchange for ~1 year prior to neutron data collection.

**X-ray and neutron data collection**. A Rigaku HighFlux HomeLab instrument equipped with a MicroMax-007 HF X-ray generator and Osmic VariMax optics was used to collect a room-temperature X-ray crystallographic data set. The X-ray instrument houses an R-Axis IV++ image plate detector. The HKL3000 software suite[35] was utilized for the diffraction data integration, and scaling. The room-temperature X-ray structures of low-pH AAT and AAT-α-methylaspartate were solved by molecular replacement using phasing information from PDB 5TOQ and 1AJS, respectively[11, 12]. The X-ray structures were refined using SHELX-2016[36]. The X-ray diffraction experimental and refinement statistics are provided in Table 1.

Quasi-Laue neutron data to 2.2 Å resolution were collected from the 0.65 mm³ AAT crystal at pH 7.5 in complex with α-methylaspartate. The crystal diffraction quality was tested and preliminary data were collected on the IMAGINE[37] instrument located at the High Flux Isotope Reactor (Oak Ridge National Laboratory). The diffraction was considered worthy of full data collection. The complete data set was collected at room temperature on the LADI-III beamline at the Institut Laue Langevin[38]. Seventeen images were collected from three manually set crystal orientations with ~21 h per neutron diffraction frame. For each diffraction image, the crystal was held stationary at different φ settings. The neutron data were processed using the Daresbury Laboratory LAUE suite program LAUEGEN, modified to account for the cylindrical geometry of the detector[39, 40]. The program LSCALE[41] was used to determine the wavelength-normalization curve using the intensities of symmetry-equivalent reflections measured at different wavelengths. No explicit absorption corrections were applied. These data were then merged in SCALA[42]. The neutron diffraction experimental and joint refinement statistics are given in Table 1.

**Joint X-ray/neutron structure refinement**. The joint XN structure of the AAT-α-methylaspartate complex at pH 7.5 was refined using nCNS[43] and manipulated in Coot[44]. Initial rigid-body refinement was followed by several cycles of atomic position, atomic displacement parameter and occupancy refinement. The structures were checked in Coot for the correctness of side-chain conformations, including orientations of OD and ND groups, and water molecule orientations based on the initial $F_O−F_C$ difference nuclear scattering length density maps. The $2F_O−F_C$ and $F_O−F_C$ nuclear scattering length density maps were then examined to ensure the correct orientation of hydroxy groups and protonation states of the enzyme residues. The protonation states of disordered side chains could not be assigned directly from the neutron scattering length density maps. All water molecules were refined as $D_2O$. During the joint XN refinement, water oxygens were first positioned at their corresponding electron density peaks, and then the $D_2O$ positions were adjusted according to the observed nuclear scattering length density. Deuterated AAT was purified and crystallized using hydrogenated reagents in $H_2O$, as deuterated reagents were cost prohibitive. Exchangeable positions are expected to be populated with protons and non-accessible positions to remain deuterated during the back exchange. Overall, the level of H/D exchange was ~85% deuteration and therefore all positions were refined for H and D. To accomplish the refinement, all H positions in AAT were modeled as D and then the occupancies of the D atoms were refined individually within the range of −0.56 to 1.00 (the scattering length of H is −0.56 times the scattering length of D). If the occupancy of the D is refined in the range of −0.56 to 0.00 it is considered non-exchanged, 0.00–0.6 is partially exchanged, and 0.6–1.0 is fully exchanged. Before depositing the final structures in the PDB, a script was used to convert a record for the coordinates of D atoms into two records corresponding to H and D atoms partially occupying the same site, with positive partial occupancies that sum to unity.

**Quantum chemical calculations**. The geometries of each model were optimized at the SMD/B3PW91/Def2-TZVPP[45, 46] level of theory. The B3PW91 functional was used in this study because it reproduced the geometries observed in the X-ray structures. The SMD solvent model[47] with a dielectric constant of 4.0 was used for

the geometry optimizations. Tight SCF and optimization convergence criteria were used. Following geometry optimization, vibrational frequencies were computed at the same level of theory to ensure that each geometry corresponds to a local minimum. The internal aldimine models were truncated at Cβ of K258, and the phosphate group was removed and replaced by a methyl group (Supplementary Data 1–4). Natural bond orbital (NBO) analysis and second-order perturbative orbital interaction energy analysis were performed with the program NBO 6.0[48]. For the NBO analysis only, the C3–C4–C4′-$N_{SB}$ torsion angle was constrained during the optimizations of the models of the internal aldimine at pH 7.5 (deprotonated $N_{SB}$) and pH 4.0 (protonated SP), and the Kohn−Sham density matrix was used for the NBO transformation (Supplementary Data 5, 6). All calculations were performed with Gaussian 09[49].

**Data availability**. Coordinates and structure factors have been deposited in the Protein Data Bank with accession codes 5VJZ and 5VK7. Other data are available from the corresponding authors upon reasonable request.

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

## Acknowledgements

The Office of Biological and Environmental Research supported research at ORNL's Center for Structural Molecular Biology (CSMB) using facilities supported by the Scientific User Facilities Division, Office of Basic Energy Sciences, US Department of Energy. This work used resources of the Compute and Data Environment for Science (CADES) at ORNL, which is managed by UT-Battelle, LLC for the U.S. Department of Energy under Contract No. DE-AC05-00OR22725. This research at ORNL's High Flux Isotope Reactor (IMAGINE beamline) was sponsored by the Scientific User Facilities Division, Office of Basic Energy Sciences, U.S. Department of Energy. The authors thank Institut Laue Langevin (beamline LADI-III) for neutron beam time. This research was supported in part by a grant from The Center for the Advancement of Science in Space, Inc. (CASIS NASA Award No. GA-2013-117, T.C.M Co-I and GA-2017-251, T.C.M. P.I.). We thank Prof. Michael Toney for insightful conversations during the manuscript preparation. S.D. was supported by CASIS, the University of Toledo, and the ORNL GO! program (Award No. N-125688-01, T.C.M. PI, A.K. ORNL PI).

## Author contributions

S.D., T.C.M., and A.Y.K. conceived and coordinated the study. S.D., T.C.M., and A.Y.K. wrote the paper. S.D., R.C.J., and J.M.P. contributed to the computational analysis. S.D., T.C.M., and A.Y.K. performed the structural analysis. M.P.B. and D.A.K, conducted the neutron data collection. S.D., O.G. and K.L.W, contributed to deuterated protein preparation. The manuscript was written through the contributions of all authors.

## Additional information

**Competing interests:** The authors declare no competing financial interests.

