## [Peer Review file · Nature Communications]

Reviewers' comments:

Reviewer #1 (Remarks to the Author):

The authors present new impressive insights into the aspartate aminotransferase system through detailed neutron and X-ray structural analysis, supported by quantum chemical calculations. All of the findings are well supported and of high interest to the biochemistry of the pyridoxal phosphate cofactor. I find particularly relevant the dismissal of the ground state destabilisation hypothesis. Many works try to highlight strain effects in enzymes, pinning reactivity to destabilisation of reactants/cofactors without a thorough verification of such hypothesis. Quantum chemical calculations are often times needed to distinguish between steric hindrance and electronic effects. This is successfully done in this manuscript.

Overall, I find the procedures and the discussion sound. The application of NBO for the analysis of the electronic structure, although I am often times sceptical about the method, is well warranted in this case. It is a very significant work, which I believe will have a strong impact in the community. I support publication as is.

Reviewer #2 (Remarks to the Author):

This manuscript describes a detailed structure of an enzyme that is very complicated. Since all of the chemistry resides in the cofactor, the protonation states of the various ionizable parts of the cofactor are presumably very important for the determination of the type of reaction catalyzed. Consequently, this is an important structure to begin to make sense of these protonation states. However, the presentation of the results needs better interpretation taking into consideration what is known about this protein (pKa's, other structures of which there are many from X-ray data).

For instance, I think a decision has to be made about what is presented here: models based on crystallographic and spectroscopic data, or interpretation based on biochemical intuition. They are not the same and may be inconsistent.

Toward a better understanding of the active site a number of issues might be addressed.

Figures should be made to illustrate the point being made.

Figure 1: it is impossible to see anything. Is it possible to get the same view for both, and possibly a better one than that given?

Figure S1: needs a figure legend

Figure 2: what is shown are not coordinates, they are models

2B: there is no evidence for the proton from the amino acid being the one transferred to the Schiff base (see comment above).

Figure S2: how were the directions of the water molecules determined? Are these waters observed in the X-ray derived model?

Figure 3: what is shown are not coordinates, they are models. A and C should be in stereo

Figure 4: the result from the neutron structure is that the Schiff base is unprotonated, but in this figure it is shown as protonated. What is known about the protonation state of lysine (shown here as neutral)? Perhaps the figure could be labeled as a "suggested mechanism".

Figure 5: what is shown are not coordinates, they are models. A great interpretation is given, but it needs clarification.

There are several structures of AspAT's from different organisms, in both the external and internal

aldimine forms. Are there torsional differences seen in them also for the O3' to Schiff base configuration?

Reviewer #3 (Remarks to the Author):

This paper represents a substantial accomplishment – the first neutron diffraction crystal structure of a PLP dependent enzyme. PLP-dependent enzymes make up 4% of catalogued enzymes and catalyze a vast array of amino acid transformations; to truly understand this rich chemistry and the acid-base mechanism of these enzymes requires that hydrogen atoms be located. This paper is the first example of such an accomplishment – demonstrating direct visualization of critical hydrogen bonds in the active site(s) of aspartate aminotransferase using neutron diffraction. Such structures hold great promise for insight into the mechanism of action and this work delivers on this promise. While a few of the results merely confirmed the protonation states inferred by earlier X-ray structures (still an accomplishment), several of the results were surprising and offer unique perspectives on the mechanism. These include the change in protonation states on H189, which is found to be neutral in the internal aldimine, but becomes protonated in the external aldimine. The resultant extra positive charge is an additional counterbalance to the negative charge on Ca in the ensuing carbanion. As well, in the external aldimine, a deuterium was observed midway between the Schiff base nitrogen and the substrate carboxylate. This unique structure is similar to the equilibrium proposed in reference 8 and my only substantial scientific question on this work is how well the authors can differentiate between the proposed low-barrier hydrogen bond and tautomeric exchange (this seems like a question that solid-state NMR might help address at a later point). Otherwise, this paper is ready and should be accepted for publication.

Reviewer #4 (Remarks to the Author):

In the manuscript entitled 'Direct visualization of critical hydrogen atoms in a pyridoxal 5'-phosphate enzyme', Dajnowicz and co-workers report for the first time a neutron crystal structure of a PLP-dependent enzyme, an aspartate aminotransferase (AAT). Taking advantage of a particularity of the crystal, each of the two monomers in the asymmetric unit present different states of the AAT reaction: the internal and external aldimines. Hence, the neutron crystal structure reveals the protonation states of key residues in the active site, as well as of the Schiff Base and other atoms of aldimine. Unexpectedly, in the external aldimine, protonation of Nsb is observed, and not of O3' as previously supposed. The authors also report a low pH (~4.0) X-ray structure, to assess the structural changes of internal aldimine upon protonation of Nsb. Finally, DFT calculations have been used to understand the origin of the out-of-plane conformation of the Schiff base in the internal aldimine. Results presented suggest that such geometry originates from intramolecular electronic forces, and not from strain caused by the side chain of K258.

While all these results are of capital importance to decipher the exact chemistry of such important class of enzymes, the manuscript in its actual form cannot be accepted without important comments to be addressed.

Major comments:

In the Results part, the neutron structure is well described and there is no major comment on this section (see below for details).

1) Regarding the low-pH structure section, there is a striking difference between the angles you report for the external aldimine at pH 7.5 (-28 deg) and pH 4 (26 deg). The external aldimine is not the focus of the paper, but such a striking difference should be explained (even in Supplemental Information if space is limited). Would that come from the protonation of the

substrate carboxylate?

2) For the DFT calculations, the justification for a truncated version of the PLP is the computational cost (in the Methods section). While legitimate for the phosphate group, adding the main chain atoms for the lysine does not seem that costly (N-Ca-C for example). Could you please comment on this point?

3) Also, the PLP pyridinium ring is Pi-stacked to a Trp residue. How this interaction would affect the orbitals of the conjugated PLP, and the conclusions of your calculations ?

4) While you give numerical values for the hyperconjugative interactions (5.2 and 12.3 kcal/mol), none is given for lone-pair repulsion. Is it possible to calculate one? If so, please provide it to strengthen your statement that "favorable conjugative and hyperconjugative interactions offsets the disruption of conjugation caused by lone pair repulsion). Please, also provide numerical values when you state that "hyper conjugative interactions are significantly decreased in the protonated SB model". If increases/decreases of energies are discussed, please provide all values.

5) You also mention a rather small deviation between the (C3-C4-C4'-Nsb) angle in the pH 7.5 neutron structure (46 deg) and in the unconstrained optimized geometry of internal aldimine (42 deg), how good is agreement between the Lys 258 Chi angles in the neutron structure and the computed model ?

6) Finally, at pH4.0, with a protonated Nsb, the angle is 22 deg, compared to 0 deg in the computed model. The agreement is not as good as before, could you comment on this please? Is it possible that the out of plane configuration is maintained by the intramolecular electronic forces, but a completely planar configuration is impossible because of the lysine / active site constraints?

While the first part of discussion is well written, it should not end, in my opinion, with a summary of the study findings. Indeed, some of these findings have already been discussed earlier in the manuscript, and it is not necessary to mention them again.

You could still follow how your findings apply to the transamination reaction, but with more in depth discussion at each point.

Comments regarding the discussion:

- The low-barrier hydrogen bond seems to be expected in the PLP mechanism between Nsb and O3' of PLP, a reference should point to studies mentioning it at the end of the paragraph in which authors state: "the LBHB would be expected to have formed between the between Nsb and O3' of PLP"
- Authors mentioned QM/MM calculations which investigated the tautomeric equilibrium between the Nsb and O3' in AAT, but do not discuss it. If this study is mentioned, some results or conclusions should be discussed in the light of the new findings of this manuscript.

In the paragraph mentioning the NMR study on the TRP-synthase, the comparison is interesting as they suggest that the protonation of N1-PLP can modulate type of chemistries. 2 comments in this paragraph:

- Why mentioning the different exchange rates of the D atom between the AAT internal and external amides? It does not seem to add anything in the comparison.
- Is there any clear differences in their respective active sites between the two enzymes that could account for the difference in protonation state ?

Minor comments:

- To ease reading of this manuscript, please provide an additional Figure of the atom labeling and

torsional angle convention for the PLP. Also, reporting O3' and Nsb in all chemical representations of PLP could be useful for the reader.

- You could mention that only a fraction of the deuterium have been omitted for Fo-Fc map calculation. Otherwise it can be confusing for the reader to have peaks only on some deuterium atoms represented in the figure.
- You mention PLP all along the manuscript for both internal and external aldimines, while using PLA for external aldimine in the figures (without any information on the meaning of PLA).
- In the second Results paragraph, when you specify that pKa estimations of Nsb vary, please give a range of values.
- Figure 2B is not cited
- At the end of the fourth paragraph in the Results section: you mention a stronger and less mobile HBond of PLP but you do not specify the partner.
- Please rephrase the first sentence of the following paragraph: "The H bond network ... changes protonation state"
- At the end of the Results section, the last sentence referring to the software used should be removed. A similar sentence is already in the methods section. Different references have been used though, please correct.
- In the Low-pH X-ray structure paragraph - Figure 5 is mentioned - Should be Figure 6.
- Figure 7: Panel A and Panel B should be replaced by (A) and (B)- (C) should be added.
- Figure S1: You mention a small domain but do not properly introduce or describe it.

Reviewers' comments:

Reviewer #1 (Remarks to the Author):

The authors present new impressive insights into the aspartate aminotransferase system through detailed neutron and X-ray structural analysis, supported by quantum chemical calculations. All of the findings are well supported and of high interest to the biochemistry of the pyridoxal phosphate cofactor. I find particularly relevant the dismissal of the ground state destabilisation hypothesis. Many works try to highlight strain effects in enzymes, pinning reactivity to destabilisation of reactants/cofactors without a thorough verification of such hypothesis. Quantum chemical calculations are often times needed to distinguish between steric hindrance and electronic effects. This is successfully done in this manuscript.

Overall, I find the procedures and the discussion sound. The application of NBO for the analysis of the electronic structure, although I am often times skeptical about the method, is well warranted in this case. It is a very significant work, which I believe will have a strong impact in the community. I support publication as is.

We greatly appreciate the Reviewer's enthusiasm toward our manuscript and the time and effort of the Reviewer.

Reviewer #2 (Remarks to the Author):

This manuscript describes a detailed structure of an enzyme that is very complicated. Since all of the chemistry resides in the cofactor, the protonation states of the various ionizable parts of the cofactor are presumable very important for the determination of the type of reaction catalyzed. Consequently, this is an important structure to begin to make sense of these protonation states. However, the presentation of the results needs better interpretation taking into consideration what is known about this protein (pKa's, other structures of which there are many from X-ray data).

For instance, I think a decision has to be made about what is presented here: models based on crystallographic and spectroscopic data, or interpretation based on biochemical intuition. They are not the same and may be inconsistent.

Toward a better understanding of the active site a number of issues might be addressed.

Reply: Yes, we agree that models based on crystallographic and spectroscopic data, or interpretation based on biochemical intuition, are not the same. This comment is addressed in the manuscript. Specifically, we removed the biochemical intuition comments.

Figures should be made to illustrate the point being made.

Figure 1: it is impossible to see anything. Is it possible to get the same view for both, and possibly a better one than that given?

Reply: We have improved Figure 1 as suggested by the reviewer. Specifically, the blow-up images of the active sites (in sticks) are made larger to clearly show the important active site residues. In the external aldimine state the small domain is closed, and the view that we present is the best we can make. We have also updated the figure caption to refer to residues Arg292 and Arg386, which move into the active site of the external aldimine due to the small domain motion and are now in contact with the cofactor/substrate, but these two residues are farther away in the active site of the internal aldimine.

Figure S1: needs a figure legend

Reply: We have expanded the figure legend and also labeled the crystal contacts shown within the image for Figure S1.

Figure 2: what is shown are not coordinates, they are models

Reply: Yes, we agree with the reviewer. What is shown are the refined structural models based on the experimental diffraction data. We corrected the figure legend in the manuscript.

2B: there is no evidence for the proton from the amino acid being the one transferred to the Schiff base (see comment above).

Reply: Yes, there is no evidence for the proton from the amino acid being the one transferred to the Schiff base. However, we propose this pathway based on the neutron structure. This pathway is chemically reasonable and feasible. The figure caption now states, “(B) Michaelis complex and proposed substrate activation with the proton transferred during substrate binding shown in red. A direct or indirect proton transfer mechanism is plausible.”

Figure S2: how were the directions of the water molecules determined? Are these waters observed in the X-ray derived model?

Reply: Yes, the waters were observed in the X-ray data, but the positions of the hydrogen atoms were not determined. The orientations of the D₂O molecules were determined based on the nuclear scattering length density maps. Importantly, the reported neutron structure is refined using a joint X-ray/neutron refinement approach, in which one model containing all the atoms is refined concurrently against both datasets, as described in the Methods section.

The method section also states: “All water molecules were refined as D₂O. Initially, water oxygens were positioned using their electron density peaks, and then were shifted slightly in accordance with the nuclear scattering length density.”

Figure 3: what is shown are not coordinates, they are models. A and C should be in stereo

Reply: As is the case for Figure 2, we corrected the caption in the manuscript. The stereo view does not help the quality of the figure, and also it would take considerably more space in the manuscript. Also, our experience is that many readers are not able to visualize stereo images. Thus, we believe that stereo figures would not provide significant improvement of the structure presentation.

Figure 4: the result from the neutron structure is that the Schiff base is unprotonated, but in this figure it is shown as protonated. What is known about the protonation state of lysine (shown here as neutral)? Perhaps the figure could be labeled as a “suggested mechanism”.

Reply: The reviewer, perhaps, refers to the internal aldimine, where indeed the Schiff base is not protonated. To make Figure 4 not redundant with Figure 2, we started the reaction scheme from the external aldimine. In the external aldimine we observe protonation of the Schiff base in our neutron structure. We changed the caption to “Proposed mechanism”.

Figure 5: what is shown are not coordinates, they are models. A great interpretation is given, but it needs clarification.

Reply: The caption has been changed to, “Figure 5. Extended hydrogen bond network near N1-PLP in the internal aldimine (A) and external aldimine (B). (C) Proposed Grotthüss proton hopping mechanism, leading to the protonation of H189 during the formation of the external aldimine.”

There are several structures of AspAT's from different organisms, in both the external and internal aldimine forms. Are there torsional differences seen in them also for the O3' to Schiff base configuration?

Reply: Yes, the manuscript now states, "The out-of-plane geometry for the SB observed in our neutron structure was also identified in other aminotransferase enzymes, with the torsion angle range of 43-96°¹³."

Reviewer #3 (Remarks to the Author):

This paper represents a substantial accomplishment – the first neutron diffraction crystal structure of a PLP dependent enzyme. PLP-dependent enzymes make up 4% of catalogued enzymes and catalyze a vast array of amino acid transformations; to truly understand this rich chemistry and the acid-base mechanism of these enzymes requires that hydrogen atoms be located. This paper is the first example of such an accomplishment – demonstrating direct visualization of critical hydrogen bonds in the active site(s) of aspartate aminotransferase using neutron diffraction. Such structures hold great promise for insight into the mechanism of action and this work delivers on this promise. While a few of the results merely confirmed the protonation states inferred by earlier X-ray structures (still an accomplishment), several of the results were surprising and offer unique perspectives on the mechanism. These include the change in protonation states on H189, which is found to be neutral in the internal aldimine, but becomes protonated in the external aldimine. The resultant extra positive charge is an additional counterbalance to the negative charge on C α in the ensuing carbanion. As well, in the external aldimine, a deuterium was observed midway between the Schiff base nitrogen and the substrate carboxylate. This unique structure is similar to the equilibrium proposed in reference 8 and my only substantial scientific question on this work is how well the authors can differentiate between the proposed low-barrier hydrogen bond and tautomeric exchange (this seems like a question that solid-state NMR might help address at a later point). Otherwise, this paper is ready and should be accepted for publication.

Reply: We greatly appreciate the Reviewer's interest and passion toward neutron crystallography. To answer the question concerning LBHBs and tautomerism, we emphasize that, indeed, neutron crystallography can differentiate between the two phenomena. If there was tautomeric exchange, let's say 50/50, then we would observe two peaks in the nuclear density map. One peak would correspond to the N-D tautomer and the second peak would correspond to the O-D tautomer. Thus, this situation essentially corresponds to a hydrogen with a small but significant energy barrier. In an LBHB, that energy barrier is about the same height as the vibrational energy levels for the hydrogen. Thus, the proton can move almost freely between N and O atoms. Quantum mechanics then requires that the highest probability for the proton position is halfway between N and O. That is exactly what we observed in the neutron structure.

Reviewer #4 (Remarks to the Author):

In the manuscript entitled ‘Direct visualization of critical hydrogen atoms in a pyridoxal 5’-phosphate enzyme’, Dajnowicz and co-workers report for the first time a neutron crystal structure of a PLP-dependent enzyme, an aspartate aminotransferase (AAT). Taking advantage of a particularity of the crystal, each of the two monomers in the asymmetric unit present different states of the AAT reaction: the internal and external aldimines. Hence, the neutron crystal structure reveals the protonation states of key residues in the active site, as well as of the Schiff Base and other atoms of aldimine. Unexpectedly, in the external aldimine, protonation of Nsb is observed, and not of O3’ as previously supposed. The authors also report a low pH (~4.0) X-ray structure, to assess the structural changes of internal aldimine upon protonation of Nsb. Finally, DFT calculations have been used to understand the origin of the out-of-plane conformation of the Schiff base in the internal aldimine. Results presented suggest that such geometry originates from intramolecular electronic forces, and not from strain caused by the side chain of K258.

While all these results are of capital importance to decipher the exact chemistry of such important class of enzymes, the manuscript in its actual form cannot be accepted without important comments to be addressed.

Major comments:

In the Results part, the neutron structure is well described and there is no major comment on this section (see below for details).

1) Regarding the low-pH structure section, there is a striking difference between the angles you report for the external aldimine at pH 7.5 (-28 deg) and pH 4 (26 deg). The external aldimine is not the focus of the paper, but such a striking difference should be explained (even in Supplemental Information if space is limited). Would that come from the protonation of the substrate carboxylate?

Reply: We have found and corrected two typographical errors in the **Method Section** for the **Low-pH X-ray structure**. The pH 4 structure is that of the internal aldimine state only. To make this point clearer, the manuscript now states, “To probe structural changes that occur upon internal aldimine SB protonation, we obtained a low-pH X-ray structure of AAT in the internal aldimine state. In the other monomer of the pH 4.0 structure (chain A with restricted motion), the corresponding torsion angle and the (Y225)O[⋯]O(O3’) distance are 26° and 3.0 Å, respectively.”

2) For the DFT calculations, the justification for a truncated version of the PLP is the computational cost (in the Methods section). While legitimate for the phosphate group, adding the main chain atoms for the lysine does not seem that costly (N-Ca-C for example). Could you please comment on this point?

Reply: Yes, we agree with the reviewer that adding the main chain atoms will not significantly increase the computational cost. If the main chain atoms were added, we would need to fix their positions during the geometry optimizations to their corresponding crystallographic positions. Our intention, and the goal, was to run unconstrained geometry optimizations on the internal aldimine models to ensure that only electronic effects were probed; thus, we truncated the models at C β . We have indeed shown that intramolecular electronic effects have a very significant influence on the co-planarity of the PLP and SB. In addition, adding the main chain atoms would not significantly influence the intramolecular conjugation, hyperconjugation, and lone pair repulsion effects in the PLP and SB as the main chain atoms are separated from SB by several CH₂ groups. Still, when we computed these models with the main chain atoms added only small (< 2.5 kcal/mol) deviations are observed for the orbital donor-acceptor interactions.

3) Also, the PLP pyridinium ring is Pi-stacked to a Trp residue. How this interaction would affect the orbitals of the conjugated PLP, and the conclusions of your calculations?

Reply: Yes, it is possible that π - π stacking of PLP with Trp could attenuate the degree of conjugation to some extent. However, the main focus of the present study is on the role of cofactor protonation in directly influencing reactivity. We view protonation and intramolecular orbital interactions as first-order effects, whereas noncovalent interactions with nearby residues are important second-order effects. Evidence for this viewpoint is the reasonable reproduction of the crystallographic geometries in minimal models that lack π - π stacking from Trp140. The DFT calculations show that intramolecular orbital interactions between the Schiff base and pyridine ring contribute significantly to conjugation, which influences non-coplanarity.

4) While you give numerical values for the hyperconjugative interactions (5.2 and 12.3 kcal/mol), none is given for lone-pair repulsion. Is it possible to calculate one? If so, please provide it to strengthen your statement that “favorable conjugative and hyperconjugative interactions offsets the disruption of conjugation caused by lone pair repulsion). Please, also provide numerical values when you state that “hyper conjugative interactions are significantly decreased in the protonated SB model”. If increases/decreases of energies are discussed, please provide all values.

Reply: Unfortunately, the NBO program cannot compute the numerical values for lone-pair repulsion. The NBO program has been designed to compute numerical values for electron donor-

acceptor interactions. We now added in the text, “In the protonated SB model, hyperconjugative interactions between N_{SB}-C4' and C3-C4 significantly decrease, to < 0.5 kcal/mol.”

5) You also mention a rather small deviation between the (C3-C4-C4'-Nsb) angle in the pH 7.5 neutron structure (46 deg) and in the unconstrained optimized geometry of internal aldimine (42 deg), how good is agreement between the Lys 258 Chi angles in the neutron structure and the computed model?

Reply: We observed only minor deviations (~5 deg or less) for the chi angles in the DFT models. For instance, the experimental χ_3 angle for Lys258 Chi is 177 deg in the neutron structure and 179 deg in the DFT model. Thus, the agreement between computation and experiment is very good.

6) Finally, at pH4.0, with a protonated Nsb, the angle is 22 deg, compared to 0 deg in the computed model. The agreement is not as good as before, could you comment on this please? Is it possible that the out of plane configuration is maintained by the intramolecular electronic forces, but a completely planar configuration is impossible because of the lysine / active site constraints?

Reply: Yes, it is possible that geometric restraints in the form of intermolecular interactions contribute to the out-of-plane geometry. However, the DFT calculations show that intramolecular orbital interactions between the Schiff base and phenolic oxygen also contribute. There are parallels with the reviewer's question in comment 3 above. To make sure that this point is clear in the manuscript we added, “This finding is consistent with the reduced torsion angle of 22° in the pH 4.0 X-ray structure, in which we expect N_{SB} to be protonated. The deviation in the torsion angle between the DFT model and the low-pH structure can be attributed to geometric restraints imposed by active site residues that are not present in the DFT models. For example, the π - π stacking interaction between the pyridine ring of PLP and W140 were excluded in the simplified model. Nevertheless, the intramolecular orbital interactions can be considered primary (first-order) effects, whereas noncovalent interactions with nearby residues are second-order.”

While the first part of discussion is well written, it should not end, in my opinion, with a summary of the study findings. Indeed, some of these findings have already been discussed earlier in the manuscript, and it is not necessary to mention them again. You could still follow how your findings apply to the transamination reaction, but with more in depth discussion at each point.

Reply: We believe that this summary displays the most significant findings in a very concise easy to grasp form, and how they relate to the AAT function. Therefore, we believe that this summary will be very helpful to readers as it emphasizes the most important conclusions from our study. We reworded bullet (1).

Comments regarding the discussion:

- The low-barrier hydrogen bond seems to be expected in the PLP mechanism between Nsb and O3' of PLP, a reference should point to studies mentioning it at the end of the paragraph in which authors state: "the LBHB would be expected to have formed between the between Nsb and O3' of PLP"

Reply: In the discussion we now write, "Here, the formation of the intramolecular LBHB between the N_{SB} and carboxylate oxygen was unexpected. Historically, a double-well hydrogen bond would be expected to have formed between the N_{SB} and O3' of PLP³.

- Authors mentioned QM/MM calculations which investigated the tautomeric equilibrium between the Nsb and O3' in AAT, but do not discuss it. If this study is mentioned, some results or conclusions should be discussed in the light of the new findings of this manuscript.

Reply: And the next paragraph states: "Previous quantum mechanics/molecular mechanics (QM/MM) calculations computed that the tautomeric equilibrium between the N_{SB} and O3' in AAT is shifted towards the N_{SB} by > 7.0 kcal/mol³¹"

In the paragraph mentioning the NMR study on the TRP-synthase, the comparison is interesting as they suggest that the protonation of N1-PLP can modulate type of chemistries. 2 comments in this paragraph:

- Why mentioning the different exchange rates of the D atom between the AAT internal and external amides? It does not seem to add anything in the comparison.

- Is there any clear differences in their respective active sites between the two enzymes that could account for the difference in protonation state ?

Reply: Yes, we agree with the reviewer and removed this comparison of H/D exchange. We think that the rest of the discussion is important and interesting to point out the active site differences in AAT and tryptophan synthase. The manuscript now states, "A recent NMR study has shown that the internal aldimine of tryptophan synthase has a protonated N_{SB} and a deprotonated N1-PLP³⁴. In AAT, N1-PLP is protonated in both the internal and external

aldimine forms. AAT catalyzes transaminations, whereas tryptophan synthase performs β -eliminations. The difference in the active site local environments between AAT and tryptophan synthase is responsible for the different observed protonation profiles. Specifically in AAT, an aspartic acid residue forms a salt bridge with N1-PLP, while in tryptophan synthase a serine residue is hydrogen bonded to N1-PLP. The different protonation states observed in the two enzymes may be responsible for promoting these two specific types of chemistries and preventing side reactions.”

Minor comments:

- To ease reading of this manuscript, please provide an additional Figure of the atom labeling and torsional angle convention for the PLP. Also, reporting O3' and Nsb in all chemical representations of PLP could be useful for the reader.

Reply: This is added to Figure 6

- You could mention that only a fraction of the deuterium have been omitted for Fo-Fc map calculation. Otherwise it can be confusing for the reader to have peaks only on some deuterium atoms represented in the figure.

Reply: This is addressed in figure captions

- You mention PLP all along the manuscript for both internal and external aldimines, while using PLA for external aldimine in the figures (without any information on the meaning of PLA).

Reply: This is addressed in the figure 1 caption.

- In the second Results paragraph, when you specify that pKa estimations of Nsb vary, please give a range of values.

Reply: This is addressed in the text.

- Figure 2B is not cited

Reply: This is addressed in the text.

- At the end of the fourth paragraph in the Results section: you mention a stronger and less mobile HBond of PLP but you do not specify the partner.

Reply: This is addressed in the text.

- Please rephrase the first sentence of the following paragraph: “The H bond network ... changes protonation state”

Reply: This is addressed in the text. “The H bond network consisting of D222, H143, T139, H189, and a cluster of water molecules (Figure S2) is coupled from N1-PLP to the bulk solvent.”

- At the end of the Results section, the last sentence referring to the software used should be removed. A similar sentence is already in the methods section. Different references have been used though, please correct.

Reply: This is addressed in the text.

“All donor-acceptor orbital interactions were computed with a > 0.5 kcal/mol threshold^{27,28} (see Supplemental Information).”

- In the Low-pH X-ray structure paragraph - Figure 5 is mentioned - Should be Figure 6.

Reply: This is addressed in the text.

-Figure 7: Panel A and Panel B should be replaced by (A) and (B)- (C) should be added.

Reply: This is addressed in the figure 7 caption.

- Figure S1: You mention a small domain but do not properly introduce or describe it.

Reply: Figure S1 has been improved to address this issue.

REVIEWERS' COMMENTS:

Reviewer #2 (Remarks to the Author):

The authors have addressed most of the comments from the previous reviews and the manuscript is therefore ready to be accepted.

Reviewer #3 (Remarks to the Author):

Although I was in favor of accepting the original paper, I do find the revised submission improved – making an important paper even better. I have read the response to my comments and those of the other reviewers and believe that they have been addressed meaningfully and completely, and that this paper should be accepted as is for publication. Again, it is a substantial accomplishment.

Reviewer #4 (Remarks to the Author):

The authors have addressed my comments and modified the manuscript accordingly. Even though I am not fully convince about the bullet point listing in the discussion, I understand the authors' point of view. I now strongly recommend publication of this work and would like to congratulate the authors for this study.